Knowledge of safe handling, administration, and waste management of chemotherapeutic drugs among oncology nurses working at Khartoum Oncology Hospital, Sudan

Sargidy Abdelgayoom Alhag Warsha 1 2
Yahia Amira 3
Ahmad Mehrunnisha 3
Abdalla Adel 4 5
Khalil Suhail Naser 1
Alasiry Sharifa 3
Shaphe Mohammad Abu 6
Mir Shabir Ahmad 7
Kashoo Faizan Z. f.kashoo@mu.edu.sa 8
1 Emergency Department, Erada and Mental Health Complex , Taif , Saudi Arabia
2 Department of Nursing, Khartoom Hospital , Khartoum , Sudan
3 Department of Nursing, College of Applied Medical Sciences, Majmaah University , Majmaah , Riyadh , Saudi Arabia
4 Nursing Science Department, College of Applied Medical Sciences, Shaqra University , Shaqra , Riyadh , Saudi Arabia
5 Nursing Department, Sinnar University , Faculty of Medicine & Health Sciences , Sinnar city , Sudan
6 Department of Physical Therapy, College of Applied Medical Sciences, Jazan University , Jazan , Saudi Arabia
7 Department of Medical Laboratory Sciences, College of Applied Medical Sciences, Majmaah University , Majmaah , Riyadh , Saudi Arabia
8 Department of Physical Therapy and Health Rehabilitation, College of Applied Medical Sciences, Majmaah University , Majmaah , Riyadh , Saudi Arabia
Zhang Xin
Electronic publication date: 2022 Oct 21
Publication date: 2022
Volume: 10
Electronic Location ID: e14173
Received 2022 Jun 2; Accepted 2022 Sep 12
Copyright: ©2022 Sargidy et al.
Copyright year: 2022
Copyright holder: Sargidy et al.
License: This is an open access article distributed under the terms of the Creative Commons Attribution License, which permits unrestricted use, distribution, reproduction and adaptation in any medium and for any purpose provided that it is properly attributed. For attribution, the original author(s), title, publication source (PeerJ) and either DOI or URL of the article must be cited.
License URL: https://creativecommons.org/licenses/by/4.0/

Keywords: Nursing, Oncology, Drug adminstration, Knowledge, Safe handling, Chemotherapy, Cancer therapy

Funding: the Deanship of Scientific Research at Majmaah University under project number R-2022-286 The authors were supported by the Deanship of Scientific Research at Majmaah University under project number: R-2022-286. The funders had no role in study design, data collection and analysis, decision to publish, or preparation of the manuscript.

==============================
Introduction

Knowledge of chemotherapeutic drug (CD) handling, administration, and waste disposal are important among nurses involved in cancer therapy. Inadequate knowledge of the management of CD could cause environmental contamination and potential harm to patients and nurses. To assess the knowledge of safe handling, administration, and waste management of CD among oncology nurses working at Khartoum Oncology Hospital, Sudan.

Methods

A questionnaire was developed by a team of experts to assess the knowledge in three domains of oncology nursing practice (handling, administration, and disposal). The study involved 78 oncology nurses working in Khartoum Oncology Hospital in Sudan from April 2020.

Results

The mean CD knowledge score of nurses was 12.7 ± 3.9 out of 26 items in the questionnaire. For each domain, their knowledge showed poor scores related to safe handling (mean = 2.0 ± 1.5 out of eight knowledge items) and good scores for administration (mean = 6.2 ± 1.7 out of 10) and poor scores for waste disposal (mean = 4.4 ± 1.5 out of eight). Simple linear regression indicated that education level (β = 3.715, p = .008) and training (β = 0.969, p = .004) significantly predicted knowledge among nurses.

Conclusion

There is a significant need to enhance the knowledge and safe handling skills of CD among oncology nurses in Sudan. Implementation of strict guidelines to manage cytotoxic waste to reduce health risks and hospital contamination.

Introduction

Cancer is a deadly disease, accounting for nearly 10 million deaths in 2020 alone (Hulvat, 2020; Turner et al., 2020). The most common are breast, lung, colon, rectum, and prostate cancers (Boakye et al., 2021). Cancer mortality has been reduced due to advancements in medicine and early detection technology (Boakye et al., 2021). Many types of treatments are available for patients with cancer, including chemotherapy, radiotherapy, hormone therapy, immunotherapy, and surgeries (Abbas & Rehman, 2018). Chemotherapy uses drugs to stop or slow the growth of cancer cells. It is administered by various routes, including oral, intravenous, intrathecal, intraperitoneal, and intra-arterial (Tanay, 2020).

In Sudan, national population-based Cancer Registry was established in 2009 (Saeed et al., 2014). In the years 2009–2010, 6,771 new cancer cases were reported, with 3,646 (53.8%) female and 3,125 (46.2%) male cases (Saeed et al., 2014). The most commonly diagnosed among women was breast cancer, followed by leukemia, cervical, and ovarian cancer. Prostate cancer was the most common among men, followed by leukemia, lymphoma, oral, colon, and hepatocellular carcinoma. Leukemia was the most frequent cancer in children under the age of 15, followed by lymphoma and cancers of the eye, bone, kidney, and brain (Saeed et al., 2014).

 Nurses are responsible for the safe and effective administration of cancer therapy medications also known as chemotherapuetic drugs or cytotoxic durgs (CD) (Kav et al., 2008). Oncology nurses are health care professionals who are trained to administer cancer therapy medications such as chemotherapy (Neuss et al., 2016). These nurses must have in-depth knowledge and skills to handle CD and patient safety. Oncology nurses must keep their knowledge and skills current with advances in cancer treatment to reduce the risk of error and promote best practices in cancer care (Rubin et al., 2015). Oncology nurses must attend regular workshops and in-service education (Khalil et al., 2017). It was reported that nurses exposed to cytotoxic drugs are more likely to have children with learning disabilities (Martin, 2005). A large-scale study conducted among (n = 56, 213) oncology nurses in Canada reported a significant increased risk of breast cancer, rectal cancer and congenital anomalies in their offspring’s (Ratner et al., 2010). A study reports that the inadequate knowledge among nurses about extravasation of cytotoxic drugs causes severe damage to patients as well as healthcare professionals (Hussin & Razaq Ahmed, 2020). Little research has been conducted related to knowledge, safe handling, and practice of oncology nurses in Sudan. Therefore, the goal of this study was to assess the knowledge of administration and waste management of chemotherapeutic drugs by nurses who work in Oncology department at Khartoum Oncology Hospital.

Methods

Design and participants

Ethical approval for the study was obtained from the National Ribat University, Faculty of Graduate Studies and Scientific research with approval number 269930. This cross-sectional study was conducted to determine the knowledge of safe handling, administration, and waste management of chemotherapeutic drugs among oncology nurses working at Khartoum Oncology Hospital, Sudan.

Nurses who work in Oncology department at Khartoum Oncology Hospital were eligible to participate. Founded in 1967 in the state of Khartoum, Sudan, Khartoum Oncology Hospital has a well-established oncology department. It is a 172-bed hospital with approximately 200 nursing staff. We used purposive sampling to invite participants to our study. Of the 92 nurses who work in Oncology department, 78 nurses having one year of work experience in the Oncology department fulfilled the criteria for inclusion. The eligible nurses were requested to provide a written consent form. We excluded nurses on vacation (n = 10) and those in administrative positions (n = 4).

Operational definitions of variables

Knowledge, safe handling, and barriers were the dependent variables assessed through a questionnaire. Experience, education, and training of oncology nurses were the independent variables.

The knowledge of the oncology nurses was assessed in three sections. Questions in the first section were related to the preparation, use, and precautions of chemotherapeutic drugs. The questions in the second section were related to knowledge regarding safe handling and administration of chemotherapy and identifying exposure risks. The questions in the third section were related to knowledge regarding the disposal of chemotherapeutic waste and managing side effects. The questionnaire consisted of true/false questions and multiple-choice questions (MCQs) with one correct answer out of four options.

Questionnaire development and description

We used a questionnaire developed by a team of experts in oncology nursing (two professors, two clinical nurses with more than 10 years of experience, one medical doctor, and one English language expert). A first draft was created by the first author of this study based on the recommendation from the previous research (Idris, 2014; Okobia et al., 2006; Sharour, 2018). It was emailed independently to all six experts. Their recommendations were discussed and implemented by the second author of this study. The second draft was emailed again to all six experts. Any conflict was resolved among the experts through meetings and discussions. The final copy of the questionnaire was piloted among 10 oncology nurses from other states of Sudan who were not part of the current study.

The questionnaire was developed to assess the knowledge and barriers toward practice among oncology nurses. The questionnaire consisted of 35 questions divided into four sections. The first section consisted of nine questions to collect demographic data from the participants. The second section consisted of eight MCQs related to knowledge regarding chemotherapy uses and types, as well as precautions used during preparation. The third section consisted of three MCQs and seven true/false statements related to knowledge regarding the safe handling and administering of chemotherapy and identifying exposure risks. The fourth section consisted of eight MCQs related to knowledge regarding the disposal of chemotherapeutic waste and managing side effects (Appendix 1). Total points obtained from each section were converted to percentages by dividing the points obtained in each section by the total points of that section, and then multiplying the result by 100. The nurse’s knowledge was considered poor if the scoring percentage was between 0–40%, fair between 41–60%, good between 61–80%, and excellent between 81–100%.

Sample size calculation

The RaoSoft Sample Size Calculator was utilized in the calculate the sample size for this study (Raosoft, Inc. RaoSoft Sample Size Calculator, 2004, online at http://www.raosoft.com/samplesize.html as of June 1, 2017). The power level was 0.95, and the error margin was 0.05. With a total of 92 oncology nurses in Khartoum Oncology Hospital, a sample size of 75 was sufficient. In total, 92 nurses were invited to participate, and 78 completed the questionnaire, with a response rate of 81.5%.

Statistical analysis

For data analysis, the SPSS software package (SPSS Inc., Chicago, IL, USA, 2020) was used. The data were summarised by frequency and percentages. Scoring for knowlwdge of handling, disposal and adminstration of cytotoxic drugs were calculated by dividing the points obtained by the total points, and then multiplying the result by 100. One-sample chi-square and binomial tests were used to compare significance between categories of demographic data. The mean and standard deviation were used to show the dispersion of data. Linear regression analysis was used to determine the predictive value of demographic characteristics for the knowledge of oncology nurses.

Results

This study involved 78 oncology nurses and had an 81.5% response rate. The majority of participants (n = 55, 70.5%) were between the ages of 31 and 40 years. There were approximately equal numbers of male and female participants, and most had a diploma in nursing. The majority of participants reported more than 5 years of general nursing experience with insufficient training in chemotherapy during their academic years. Fewer than 20% of participants reported having undergone training in chemotherapy, and 55.1% reported having never having undergone any training in chemotherapy (Table 1).

Table 1 Demographic data (n − 78).

Variables	Frequency
(n)	Percentage
(%)	p	
Age (years)				
20–30	10	12.8	0.001*	
31–40	55	70.5	
41–50	13	16.7	
Gender				
Male	31	39.7	0.089**	
Female	47	60.3	
Education				
Diploma degree	71	91.0	0.001*	
Bachelor degree	6	7.7	
Master degree	1	1.3	
General experience (years)				
1–5	56	71.8	0.001*	
5–10	16	20.5	
10–15	1	1.3	
More than 15	5	6.4	
Chemotherapeutic experience (years)				
1–3 years	22	28.2	0.174*	
4–7 years	23	29.5	
8–11 years	11	14.1	
Above 11 years	22	28.2	
Training in chemotherapy during academic years				
Not sufficient	75	96.2	0.001**	
Not received	3	3.8	
Training in chemotherapy after graduation				
Once	15	19.2	0.001*	
Twice	17	21.8	
More than two	3	3.8	
Never	43	55.1	
chemotherapy				
Hospital has policy on safe handling of chemotherapy drugs				
yes	0	0	ND	
no	78	100	
Do you have spill kit at your workplace				
yes	0	0	ND	
no	78	100	
Notes.

* One Sample Chi-Square Test.

** One Sample Binomial Test.

ND Not determined

The mean score of knowledge on chemotherapy uses and types and the precautions used during preparation was 2.0 (poor), and the standard deviation was 1.5 (out of 8). The scores obtained for each question are given in Table 2.

Table 2 Knowledge of chemotherapy uses and types, precautions used during preparation.

Multiple-choice questions in the questionnaire	Frequency (n)	Percentage (%)	
A place to prepare chemotherapeutic drugs			
Correct answer	0	0	
Incorrect answer	78	100	
Meaning of adjuvant drugs			
Correct answer	9	11.5	
Incorrect answer	69	88.5	
Knowledge of action of cytotoxic drugs on the cell cycle phase			
Correct answer	9	11.5	
Incorrect answer	69	88.5	
Use of chemotherapy drugs			
Correct answer	27	34.6	
Incorrect answer	51	65.4	
Related to intravenous delivery of cytotoxic drugs			
Correct answer	26	33.3	
Incorrect answer	52	66.7	
Pre-administration precaution before administration of cytotoxic drugs			
Correct answer	21	26.9	
Incorrect answer	57	73.1	
The recommended sequence of the cytotoxic drugs			
Correct answer	16	20.5	
Incorrect answer	62	79.5	
What types of gloves are worn while preparing chemotherapy			
Correct answer	48	61.5	
Incorrect answer	30	38.5	

The mean score of knowledge regarding safe handling and administering of chemotherapy and identifying exposure risks was 6.2 (good), with a standard deviation of 1.7 (out of 10). The scores obtained for each question are given in Table 3.

Table 3 Knowledge regarding safe handling and administering chemotherapy, identifying exposure risks.

Questions	Frequency
(n)	Percentage
(%)	
Use of Personal protective equipment (PPE)	15	19.2	
Correct answer	63	80.8	
Incorrect answer			
Pre-medication administration time			
Correct answer	27	34.6	
Incorrect answer	51	65.4	
Related to minimize exposure to chemotherapeutic agents			
Correct answer	20	25.6	
Incorrect answer	58	74.4	
Related to the use of recommended type gloves			
Correct answer	68	87.2	
Incorrect answer	10	12.8	
Statement related to labeling the chemotherapy drug			
Correct answer	61	78.2	
Incorrect answer	17	21.8	
Related to safe handling of cytotoxic drugs			
Correct answer	51	65.4	
Incorrect answer	27	34.6	
Chemotherapy can enter the body through inhalation			
Correct answer	30	38.5	
Incorrect answer	48	61.5	
Chemotherapy can more easily enter the body through damaged skin.			
Correct answer	28	35.9	
Incorrect answer	50	64.1	
Chemotherapy tablets can be chewed, cut, or crush			
Correct answer	72	92.3	
Incorrect answer	6	7.7	
Administering/handling chemotherapy is no different from administering/ handling intravenous antibiotics.			
Correct answer	73	93.6	
Incorrect answer	5	6.4	

The mean score of knowledge regarding the disposal of chemotherapeutic waste and managing side effects was 4.4 (fair), with a standard deviation of 1.5 (out of 8). The scores obtained for each question are given in Table 4.

Table 4 Related to knowledge regarding disposal of chemotherapeutic waste and managing side effects.

Questions	Frequency
(n)	Percentage
(%)	
Related to disposal of Cytotoxic waste			
Correct answer	27	34.6	
Incorrect answer	51	65.4	
Management of spillage of chemotherapy drugs			
Correct answer	23	29.5	
Incorrect answer	55	70.5	
Scenario related to extravasation chemotherapy drugs			
Correct answer	30	38.5	
Incorrect answer	48	61.5	
Actions taken during allergic reaction to chemotherapy drug			
Correct answer	22	28.2	
Incorrect answer	56	71.8	
Knowledge of side effects of chemotherapy			
Correct answer	43	55.1	
Incorrect answer	35	44.9	
Related to re-use of gown after handing cytotoxic drugs			
Correct answer	77	98.7	
Incorrect answer	1	1.3	
Related to management of general and cytotoxic waste management			
Correct answer	61	78.2	
Incorrect answer	17	21.8	
Related to education of patient about side effects of cytotoxic drugs.			
Correct answer	63	80.8	
Incorrect answer	15	19.2	

The mean of the combined knowledge scores for all the sections was 12.7 (fair), with a standard deviation of 3.9 (out of 27).

A simple linear regression was carried out to test whether demographic variables significantly predicted knowledge of CD among nurses. The results of the regression indicated that the model explained 16.3% of the variance and that the model was significant, F (7, 70) = 3.150 (p = .006). Nursing education level and chemotherapy training significantly predicted chemotherapeutic knowledge among nurses (β = 3.715, p = 0.008 and β = 0.969, p = 0.004, respectively). Participants with bachelor’s degrees had a mean knowledge score of 19.67 (SD = 4.59), compared to a mean of 12.17 (SD = 3.334) for nurses with diplomas (Table 5).

Table 5 Linear regression analysis between demographic variables and knowledge as dependent variable.

Demographic variables	Unstandardized coefficients	Standardized coefficients	t	Sig.	95.0% Confidence Interval for B	Correlations	Collinearity statistics	
	B	Std. error	Beta			Lower bound	Upper bound	Zero- order	Partial	Part	Tolerance	VIF	
Age	−0.446	1.078	−0.062	−0.414	0.680	−2.597	1.705	−0.074	−0.049	−0.043	0.487	2.051	
Gender	−0.240	0.857	−0.030	−0.280	0.780	−1.948	1.469	−0.030	−0.033	−0.029	0.946	1.057	
Education	3.715	1.355	0.326	2.743	0.008	1.013	6.417	0.401	0.311	0.286	0.770	1.298	
General nursing experiences	0.277	0.633	0.057	0.437	0.663	−0.986	1.540	0.199	0.052	0.046	0.634	1.578	
Experience in chemotherapy administration	0.333	0.530	0.100	0.629	0.531	−0.723	1.390	−0.193	0.075	0.066	0.432	2.315	
Training course in chemotherapy	0.969	0.386	0.307	2.508	0.014	0.198	1.739	0.362	0.287	0.261	0.726	1.377	
Notes.

a Dependent Variable: Total sum scores of knowledge.

VIF Variance inflation factor

Discussion

The purpose of this study was to evaluate the knowledge of nurses regarding the safe handling, administration, and waste disposal of CD in the oncology department. The nurses working in the oncology department at Khartoum Oncology Hospital in Sudan had poor knowledge about safe handling and cytotoxic waste disposal and fair knowledge about CD administration. The knowledge scores obtained in this study were low compared to studies conducted among oncology nurses in Egypt (Zayed et al., 2019), Nepal (Chaudhary & Karn, 2012), the United states (Callahan et al., 2016), Iran (Orujlu et al., 2016), Malaysia (Keat et al., 2013), and Cyprus (Kyprianou et al., 2010). The difference may be due to variations in standard cancer guidelines at the hospital and the availability of in-service training.

All the participants reported that they were not aware of the availability of spill kits or policy on the safe handling of CD at the hospital. Similarly, research conducted in Egypt among 50 nurses reported health hazards such as skin irritation, chest allergy, irregular menstruation, and abortion (Aziza, 2019). Further, the authors of that study reported that the nurses did not use biological safety cabinets for CD preparation and storage, as well as non-compliance with personal protective equipment (PPE) and the absence of safe handling chemotherapy guidelines (Aziza, 2019). This can be attributed to a lack of strict guidelines and regulations in hospitals.

One quarter of the oncology nurses in our study gave the correct answer to questions related to the sequence of drug administration, pre-administration, and preparation, although more than half knew about the type of gloves needed for chemotherapy. Increased usage of gloves has also been reported in other countries such as Nepal and Turkey. This similarity may be due to the availability of gloves being high compared to other PPE. The lack of knowledge about the sequence of drug administration and pre-administrative precautions could cause potential harm to patients receiving cancer therapy (Cohen & Erickson, 2006; Rosenzweig et al., 2012; Schwappach, Hochreutener & Wernli, 2010).

Only a quarter of the nurses in this study reported using PPE at all stages of chemotherapy. These results could be attributed to the limited resources and lack of training at the hospital. However, more than half of the oncology nurses had fair knowledge about the recommended chemotherapy gloves, labeling of chemotherapeutic drugs, and administration of chemotherapeutic drugs. A study among 35 Egyptian oncology nurses reported good knowledge and unsatisfactory practice (Zakaria, Alaa & Desoky, 2022). Another study conducted in Iran among 225 rotational nurses reported a high level of knowledge about cytotoxic drugs and high compliance with PPE (Ali Shahrasbi et al., 2014).

More than three quarters of oncology nurses in this study could not correctly answer the questions related to the disposal of cytotoxic waste, while nurses had good knowledge related to gown re-use, patient education, and adverse effects of chemotherapy. In line with the results of our study 203 nurses from China reported insufficient knowledge of chemotherapy (Yu et al., 2013). The authors further reported that the participants gained knowledge about chemotherapy from colleagues and in-service training (Yu et al., 2013). Another study conducted among Italian oncology nurses (n = 287) showed insufficient knowledge and incorrect beliefs about cancer pain that could affect pain management (Bernardi et al., 2007). Improper disposal of cytotoxic drugs could lead to environmental contamination and harm to healthy individuals (Padmanabhan & Barik, 2019). Other possible reasons for the improper disposal of cytotoxic waste could be lack of time and work overload. Khartoum Oncology Hospital is a renowned facility, and people come to visit this hospital from various states of Sudan.

Regression analysis showed that qualifications and training significantly predicted knowledge. The mean knowledge scores of participants with a bachelor’s degree in nursing were significantly higher than of those with a diploma. This discrepancy could be due to the difference in the training and curriculum between a diploma and a bachelor of nursing. However, surprisingly, participants with no post-academic training showed higher mean knowledge scores than those with such training. This discrepancy may be due to poor quality of training and may eventually question the credibility of this training. Robust research is needed to determine the most effective training to improve the knowledge of oncology nurses.

The majority of nurses reported a shortage of available PPE, and the rest reported a lack of PPE awareness, lack of time for it, discomfort associated with PPE, and concern that PPE may upset patients (Fig. 1). A study conducted among 353 healthcare workers from 24 Greek hospitals reported a significant proportion of the sample did not use PPE (Constantinidis et al., 2011). The authors further reported that healthcare workers were not aware of the dangers of cytotoxic drugs (Constantinidis et al., 2011). The main reason for non-compliance with PPE in our study might be limited resources at Khartoum Oncology Hospitalay eened hospital peutic drugs.

Figure 1 Barriers towards safe handling of cytotoxic drugs.

Limitations

The cross-sectional study contains reporting bias. Additionally, the sample size in this study was small and drawn from a single hospital. This potentially limits the generalizability of the study.

Conclusion

There is a significant need to enhance the knowledge and safe handling skills of CD among oncology nurses in Sudan. Implementation of strict guidelines to manage cytotoxic waste to reduce health risks and contamination.

Recommendation

Oncology nurses need urgent training to safely handle CD to improve the quality of patient care and prevent cytotoxic drug contamination.

Supplemental Information

Supplemental Information 1 Raw data

Click here for additional data file.

Supplemental Information 2 Questionnaire with answers

Click here for additional data file.

Supplemental Information 3 Questionnaire used in the study

Click here for additional data file.

Additional Information and Declarations

Competing Interests

Author Contributions

Human Ethics

Data Availability

Faizan Z. Kashoo is an Academic Editor for PeerJ.

Abdelgayoom Alhag Warsha Sargidy conceived and designed the experiments, performed the experiments, analyzed the data, prepared figures and/or tables, authored or reviewed drafts of the article, and approved the final draft.

Amira Yahia conceived and designed the experiments, performed the experiments, analyzed the data, prepared figures and/or tables, authored or reviewed drafts of the article, and approved the final draft.

Mehrunnisha Ahmad conceived and designed the experiments, performed the experiments, analyzed the data, prepared figures and/or tables, authored or reviewed drafts of the article, and approved the final draft.

Adel Abdalla conceived and designed the experiments, analyzed the data, prepared figures and/or tables, authored or reviewed drafts of the article, and approved the final draft.

Suhail Naser Khalil conceived and designed the experiments, analyzed the data, prepared figures and/or tables, authored or reviewed drafts of the article, and approved the final draft.

Sharifa Alasiry analyzed the data, prepared figures and/or tables, authored or reviewed drafts of the article, and approved the final draft.

Mohammad Abu Shaphe conceived and designed the experiments, performed the experiments, analyzed the data, prepared figures and/or tables, authored or reviewed drafts of the article, and approved the final draft.

Shabir Ahmad Mir performed the experiments, prepared figures and/or tables, authored or reviewed drafts of the article, and approved the final draft.

Faizan Z. Kashoo conceived and designed the experiments, performed the experiments, analyzed the data, prepared figures and/or tables, authored or reviewed drafts of the article, and approved the final draft.

The following information was supplied relating to ethical approvals (i.e., approving body and any reference numbers):

Ethical approval for the study was obtained from the National Ribat University, Faculty of Graduate Studies and Scientific research(Approval Number: 269930).

The following information was supplied regarding data availability:

The raw data is available in the Supplemental File.

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
