# Peer review of "Knowledge of safe handling, administration, and waste management of chemotherapeutic drugs among oncology nurses working at Khartoum Oncology Hospital, Sudan"

_PeerJ, doi:10.7717/peerj.14173_

## Round 0.1 · original submission · Major Revisions

All three reviewers gave their opinions. Please try your best to answer the questions and make corrections.

·

Basic reporting

Generally good use of English throughout, no obvious concerns with reporting language. Minor corrections to be made:
-Affiliation number 2 Khartoum is written as “Khartoom”, this is off-putting to see at the beginning of the study.
-Line 33 makes it seem as if this was the first national registry worldwide whereas I believe the authors meant Sudan’s first national registry.
-Line 47 there is an unnecessary capital T for “The”.
-Line 188 should be credibility and not “creditability”.
In the abstract, there should not be an “objective” subheading and it should be Results instead of “Result”.
There should be less use of subheadings within the Methods sections as this reads more like a thesis rather than a journal article, this should be reformatted.
In the Introduction, there should be more emphasis on the role of oncology nurses in safe handling and distribution of CDs and the risks/consequences related to this rather than most of it giving general background on cancer. This needs further expansion on why your study is important

Experimental design

No concerns regarding experimental design. Fairly limited dataset but almost full coverage of eligible nurses done.

Needs some emphasis on what gap is being filled. For example that a similar study in the region has not been done before.

Validity of the findings

Robust data analysis used with linear regression.

“A first draft was created by the first author of this study based on the 89 recommendation from the previous research.” As stated by the authors in the methods section, however this is not referenced so it is unclear which previous research.

Given that the knowledge scores are arbitrarily categorised into “poor”, “fair” etc, so it is clear to the reader there should be more reporting using the categories rather than numbers as the numbers per se do not necessarily mean much.

It should also be added to methods what the basis for categorising the scores was.



Lastly the conclusion and recommendations are very generic and need revision. This study is important and falls into the realm of clinical audit and governance and hence there should be a plan as to what is expected to be done about these worrying findings.

Additional comments

no comment

·

Basic reporting

no comment

Experimental design

no comment

Validity of the findings

no comment

Additional comments

no comment

Reviewer 3 ·

Basic reporting

English was good enough, and references also were sufficient,
the article was well structured. in terms of figure, tables and so on.

Experimental design

all was well designed and of nice handling,

Validity of the findings

I have some negative opinion concerning this point.
Mind the followings:
1/ in line 15 and 16 mentioned poor knowledge in safe handling of CD waste and in the same time mentioned fair administration (some contradiction appear here ),please clarify .
2/ In limitation section(line 199), authors told that English language was a factor of the poor knowledge result for safe handling of CD , - as known all volunteers were a university graduate , and some of them were had post graduate degree . Besides more than 70% of them were of more than 4 years experience in the chemotherapy field, So the justification of these poor knowledge due to English language insufficiency is not matching with such inputs. ( this point needs more clear justification ,please clarify).
3/ statistics needs more descriptions

---

## Round 0.2 · accepted · Accept

All three reviewers have given their opinions for revision. After the author's revision and answers to questions, two of them are willing to endorse for publication, while the other one has not responded to the invitation for review (more than 2 weeks). Overall, I think this manuscript meets the requirements for publication, and I recommend that it be accepted without delay to the author.

Reviewer 3 ·

Basic reporting

Accepted as it is

Experimental design

All was clearly corrected and know accepted

Validity of the findings

well Valid